# Preservation of lymphocyte functional fitness in perinatally-infected and treated HIV+ pediatric patients displaying sub-optimal viral control

Aaruni Khanolkar [1,2 ✉], William J. Muller [3,4], Bridget M. Simpson[1], Jillian Cerullo[1], Ruth Williams[3], Sun Bae Sowers[5], Kiana Matthews[5], Sara Mercader [5], Carole J. Hickman[5], Richard T. D'Aquila [6] & Guorong Liu[1]

## Abstract

**Background** Host–pathogen dynamics associated with HIV infection are quite distinct in children versus adults. We interrogated the functional fitness of the lymphocyte responses in two cohorts of perinatally infected HIV+ pediatric subjects with early anti-retroviral therapy (ART) initiation but divergent patterns of virologic control. We hypothesized that sub-optimal viral control would compromise immune functional fitness.

**Methods** The immune responses in the two HIV+ cohorts ($n = 6$ in each cohort) were benchmarked against the responses measured in age-range matched, uninfected healthy control subjects ($n = 11$) by utilizing tests for normality, and comparison [the Kruskal–Wallis test, and the two-tailed Mann–Whitney $U$ test (where appropriate)]. Lymphocyte responses were examined by intra-cellular cytokine secretion, degranulation assays as well as phosflow. A subset of these data were further queried by an automated clustering algorithm. Finally, we evaluated the humoral immune responses to four childhood vaccines in all three cohorts.

**Results** We demonstrate that contrary to expectations pediatric HIV+ patients with sub-optimal viral control display no significant deficits in immune functional fitness. In fact, the patients that display better virologic control lack functional Gag-specific T cell responses and compared to healthy controls they display signaling deficits and an enrichment of mitogen-stimulated CD3 negative and positive lymphocyte clusters with suppressed cytokine production.

**Conclusions** These results highlight the immune resilience in HIV+ children on ART with sub-optimal viral control. With respect to HIV+ children on ART with better viral control, our data suggest that this cohort might potentially benefit from targeted interventions that might mitigate cell-mediated immune functional quiescence.

## Plain language summary

The immune response associated with HIV infection is different in children compared to adults. Efforts aimed at developing an effective vaccine against HIV have faced multiple setbacks, which suggests that we do not yet completely understand the interaction between HIV and our immune system. However, recently published studies performed in children born with HIV have indicated that perhaps manipulating the immune response in children could be beneficial. In this study we have comprehensively evaluated the activity of white blood cells in two groups of children born with HIV that display different levels of HIV in their blood. Our results show that normal white blood cell activity is unaffected in children with relatively poor level of HIV control in their blood. These results contrast with those observed in adults where poor viral control impairs white blood cell activity.

[1] Department of Pathology, Ann and Robert H. Lurie Children's Hospital of Chicago, Chicago, IL 60611, USA. [2] Department of Pathology, Feinberg School of Medicine, Northwestern University, Chicago, IL 60611, USA. [3] Division of Infectious Diseases, Ann and Robert H. Lurie Children's Hospital of Chicago, Chicago, IL 60611, USA. [4] Department of Pediatrics, Feinberg School of Medicine, Northwestern University, Chicago, IL 60611, USA. [5] Viral Vaccine Preventable Diseases Branch, Division of Viral Diseases, National Center for Immunization & Respiratory Diseases, Centers for Disease Control and Prevention, 1600 Clifton Rd., Atlanta, GA 30333, USA. [6] Division of Infectious Diseases, Department of Internal Medicine, Feinberg School of Medicine, Northwestern University, Chicago, IL 60611, USA. ✉email: AKhanolkar@luriechildrens.org

It is estimated that at the end of 2019, about 2 million children are currently living with the Human Immunodeficiency Virus (HIV)[1]. One of the major success stories in the fight against the Acquired Immunodeficiency Syndrome (AIDS) has been the development of combination anti-retroviral therapy (ART)[2]. Timely access to ART has contributed to a dramatic reduction in the mother to child transmission rates of HIV (perinatal HIV transmission accounted for only 65 out of the ~38,000 new HIV diagnoses in the US in 2018), and 50% of HIV-infected children die by the second year of life if they do not receive ART[3–5]. In contrast to ART, efforts to develop a successful vaccine against HIV have faced multiple setbacks[6]. This likely reflects the fact that we do not yet fully understand the complex host pathogen dynamics associated with HIV infection. Hence, in order to attempt a cure for this disease we need to further enhance our understanding of the virus–host interactions, be aware of its nuances, and perhaps look beyond a one size fits all approach. Related to this point, there are some intriguing differences between adults and children infected with HIV. Between 5% and 10% of ART naive, HIV+ children are slow progressors with clearly detectable viremia but they still maintain normal, age-range associated CD4 T cell counts, and they do not harbor HLA class I alleles that have been described to mediate protective CD8 T cell responses[7–9]. Prevailing evidence indicates that there does not appear to be a good correlation between viral load, immune activation, and CD4 counts in children[7,8,10]. Furthermore, HIV-associated disease can progress quickly in ART naïve children even if they harbor HLA class I alleles deemed protective and despite mounting an early and active virus-specific CD8 T cell response (presumably by CD8 T cell-mediated cytotoxicity of infected CD4 T cells)[7–9]. ART-treated children display a more rapid rebound in CD4 T cell numbers compared to adults, and a greater proportion of naïve T cells populate the peripheral T cell repertoire following ART-mediated viral control in the pediatric age group[11–14]. Moreover, the extent to which CD4 T cell function recovers following viral suppression is greater in children than that observed in treated, aviremic adults[11,13,15,16]. Age-related changes in thymic output quite likely contribute to these differences between ART-treated adults and children[14]. Interestingly, the viral reservoir in children is not populated with escape variants to the extent observed in adults in spite of the fact that immune selection pressure induces escape variants early after infection in children[7–9,17,18]. Collectively, this has led to the speculation that manipulating the immune response in children might present better odds for a pathway to a cure against HIV. This concept is also gaining traction due to the fact that the expectation of faithful drug compliance (especially during adolescence) and lifelong adherence to ART that is initiated in childhood is perhaps unrealistic[19–21]. This realization coupled with the cost and the attendant side effects of prolonged therapy is prompting a serious re-assessment of what might constitute the most effective long-term management approach for this disease[22,23].

In this study, we explored the hypothesis that sub-optimal viral control will compromise immune functional fitness in perinatally infected HIV+ patients with early ART initiation[24,25]. We evaluated the immune functional fitness of two US based cohorts of perinatally infected HIV+ study subjects that initiated ART within the first year of life. Cohort 1 consists of individuals that displayed better virologic control compared to cohort 2 study subjects reflecting more consistent adherence, as well as a lower degree of viral resistance to ART. The results from both cohorts were benchmarked against responses measured in age-range matched, uninfected control subjects. We measured polyfunctional responses (IFNγ, TNFα, IL-2, and IL-21) and degranulation potential (CD107a) by intracellular cytokine staining assays in defined lymphocyte subsets following PMA/ionomycin stimulation, as well as following Gag-potential T cell epitope (PTE)-peptide pool treatment. Additionally, we queried our mitogen-stimulated intracellular cytokine data using unsupervised clustering analyses to uncover novel populations that clearly segregated healthy control donors from the HIV-exposed subjects. We also utilized phosflow to track the phosphoprotein signature in the lymphocyte subsets following treatment with IFNγ, IFNα, IL-2, IL-4, IL-6, IL-7, IL-10, IL-15, IL-21 and after anti-CD3 stimulation. Finally, we examined serum Ab responses to childhood vaccines against Varicella, tetanus, measles and Haemophilus influenzae b (Hib).

Our data reveal that sub-optimal viral control did not subvert immune functional fitness. On the other hand, optimal viral suppression was characterized by undetectable Gag-specific T cells, subdued lymphocyte signaling responses, and an enrichment in unique, predominantly CD3 negative lymphocyte clusters, that displayed suppressed cytokine responses.

## Methods

**Study subjects.** Perinatally infected HIV+ study subjects with early ART initiation ($n = 6$ each in cohorts 1 and 2), and age-range matched, HIV-negative control subjects ($n = 11$), were recruited between June 2018–November 2019, from the Special Infectious Diseases (SID) clinic, and the Academic General Pediatrics (AGP) clinic at the Ann and Robert H. Lurie Children's Hospital of Chicago (Supplementary Table 1). Fourteen subjects were African–American, four identified as white, and race was unknown/not reported for five subjects. In terms of ethnicity, seven study subjects identified as Hispanic/Latino. This study was carried out in accordance with the recommendations of the Human Research Protection Program Guidelines of our Institutional Review Board (IRB). This protocol was approved by our IRB (No. 2018-1541; Study Title: Evaluation of immune functional fitness in perinatally infected HIV+ study subjects with early ART initiation). All study subjects ≥18 years gave written informed consent in accordance with the Declaration of Helsinki. Study subjects between the ages of 12–17 years provided written assent and in addition we obtained written informed consent from their legal guardians. Written informed consent was obtained from the legal guardians of all study subjects below the age of 12 years. For the purpose of the measles serology assays that were performed on de-identified samples at the Centers for Disease Control and Prevention (CDC), the CDC Human Research Protections Office determined that this study was not human subjects research and exempt from the CDC Institutional Review Board (IRB) review.

**Measurement of peripheral T cell counts and viral load measurements.** We are a Clinical Laboratory Improvement Amendments (CLIA)-certified and College of American Pathologists (CAP)-accredited laboratory. The peripheral CD4 and CD8 T cell counts for the study subjects were performed using our clinically validated flow-cytometry based immunophenotyping assay that utilizes the BD Multitest[TM] 6 color TBNK Reagent with Tru-count[TM] tubes (BD Biosciences, San Jose, CA) (Supplementary Data). From 1999 to November 2008, viral load quantification was performed using the Roche Amplicor HIV-1 Monitor test (Roche Molecular Systems, Inc., Branchburg, NJ). After November 2008, the Abbott Real Time HIV-1 assay (Abbott Molecular Inc., Des Plaines, IL) was deployed to measure the viral load (Supplementary Data). For the purpose of this study a cut-off value of >200 HIV-RNA copies/ml of plasma was used to define a detectable viral load.

**Staining for intracellular cytokines and CD107a by flow cytometry using whole blood samples**. We used intracellular cytokine staining per published guidelines[26–28]. Whole blood sample aliquots were treated with medium alone or stimulated with phorbol 12-myristate 13-acetate (PMA) (10 ng/mL) and Ionomycin (Ion) (1 µg/mL), or the HIV-1 potential T cell epitopes (PTE) Gag peptide pool (2 µg/ml) (NIH AIDS Reagent Program, Germantown, MD; Catalog No. 12437) for 5 hours at 37 °C in the presence of Brefeldin A (Golgi-Plug) (BD Biosciences, San Jose, CA) [or Monensin (Golgi-Stop) (BD Biosciences, San Jose, CA) for CD107a assays]. The following MAb clones directed against the human antigens listed below were used at the manufacturer recommended doses to detect the intracellular cytokines: IFNγ (clone 4S.B3); TNFα (clone MAb11); IL-21 (clone 3A3-N2), and IL-2 (clone MQ1-17H12) (all from Biolegend, San Diego, CA). Relevant isotype control Abs (Biolegend, San Diego, CA) were used at matching doses to determine background level staining (Supplementary Table 2). For CD107a detection, the relevant isotype-control Ab (mouse IgG1, κ-PE; clone X40; BD-Biosciences, San Jose, CA) or the CD107a-PE Ab (clone H4A3; BD-Biosciences, San Jose, CA) was added at the initiation of stimulation[26,28]. MAb directed against human CD3 and CD8 were used for surface staining (BD Biosciences, San Jose, CA). Lymphocytes were gated based on light scatter characteristics followed by gating on CD3+ and CD3-negative populations. CD8+ and CD8− T cells were identified within the CD3+ population. Additional subsets were similarly delineated based on surface or cytosolic marker expression within the indicated parent population. The samples were acquired using FACS Canto-II instruments (Becton-Dickinson, Franklin Lakes, NJ) (a million total events/tube were acquired for the Gag-PTE-treated tubes) and data were analyzed using FlowJo software (version ≥ 10.6) (BD Biosciences, San Jose, CA). Each study subject was analyzed once (Supplementary Data).

**Unsupervised (automated) clustering analyses**. Cluster Identification, Characterization, and Regression (CITRUS) (Beckman Coulter, Brea, CA) is an automated clustering algorithm designed with the purpose of identifying statistically significant differences in the biological properties of cellular populations identified in multiple experimental groups[29]. CITRUS was used to compare the relative abundance of cells in each subset compared to the whole population, as well as the median expression levels of the four cytokines between the healthy control samples and cohorts 1 and 2 (Supplementary Data). For evaluating the abundance feature, we selected either CD3+ or CD3− lymphocytes as the starting population, and subsequently selected CD8, IFNγ, TNFα, IL-2 and IL-21 as the clustering channels. To evaluate the median feature, lymphocytes were selected as the starting population, and CD3 and CD8 were placed in the clustering channels, and the four cytokines (IFNγ, TNFα, IL-2 and IL-21) were listed as the statistics channels. For both the median and abundance features, we utilized the Predictive-Nearest Shrunken Centroid [Partitioning Around Medoids-R (PAMR)] association model, and an equal number of events were evaluated from each fcs file. The CITRUS run was repeated a minimum of four times for every group pairing (Control versus Cohort 1, Control versus Cohort 2, and Cohort 1 versus Cohort 2) for assessing the abundance and median cluster characterizations.

**Phosflow analyses for evaluating the phosphorylation status of signaling nodes**. We performed phosflow staining using established protocols[26,30,31]. Whole blood sample aliquots were treated for 15 min at 37 ºC with phosphate buffered saline (PBS) or the following recombinant human cytokines: IFNγ, IL-4, IL-10, IL-15

(BD Biosciences, San Jose, CA), IFNα (R&D Systems, Minneapolis, MN), IL-2 (PeproTech, Rocky Hill, NJ), IL-7 (Thermo Fisher Scientific, San Diego, CA), IL-6 and IL-21 (Biolegend, San Diego, CA). The cells were then fixed, permeabilized and stained with the MAb targeting the following molecules: anti-human human CD3, CD8 (Biolegend, San Diego, CA), phospho-STAT1 (pY701; clone 4a), phospho-STAT3 (pY705; clone 4), phospho-STAT5 (pY694; clone 47), and phospho-STAT6 (pY641; clone 18) (all from BD Biosciences, San Jose, CA). For measuring phospho-ZAP70, whole blood sample aliquots were incubated with either mouse anti-human CD3 MAb (clone UCHT1) (Biolegend, San Diego, CA) or mouse IgG1, κ (clone MOPC-21; Biolegend, San Diego, CA) for 30 min on ice, washed and then treated with goat anti-mouse IgG (clone Poly4053; Biolegend, CA) on ice for an additional 30 min before transferring the tubes to a 37 °C water bath for 5 min. The cells were then fixed, permeabilized and stained per manufacturer's instructions with an antibody targeting the phospho-ZAP70 (PY319) (clone 17A/P-ZAP70; BD Biosciences, San Jose, CA) and with a non-competing Ab clone targeting CD3 (clone SK7) and a MAb to CD8 (Biolegend, San Diego, CA). 50,000–100,000 events/tube were acquired on FACS Canto-II instruments (Becton-Dickinson, Franklin Lakes, NJ) and the data were analyzed using Cytobank software (Cytobank, Santa Clara, CA). Each study subject was analyzed once (Supplementary Data).

**Microbial serology assays**. IgG responses to tetanus toxoid (TT) and Haemophilus influenzae type b (Hib) were evaluated in duplicate with clinically validated enzyme immunoassays (EIA) that utilize the VaccZyme™ TT-IgG and the VaccZyme™ Hib-IgG kit from Binding Site (Edgbaston, Birmingham, UK) (Supplementary Data). The IgG responses to Varicella-zoster virus (VZV) were also evaluated in duplicate with a clinically validated assay that utilizes the CAPTIA™ EIA kits manufactured by Trinity Biotech (Jamestown, NY) (Supplementary Data). All three EIA assays were run on the fully automated, open DS2 platform (Dynex Technologies, Chantilly, VA). The measles virus serology assays were performed at the Viral Vaccine Preventable Diseases Branch at the Centers for Disease Control and Prevention (Atlanta, GA) (Supplementary Data). Measles immunity was defined as a serum IgG titer >120 measured by plaque reduction neutralization (PRN). A PRN titer of >120 was considered protective for measles and is equivalent to 120 mIU/ml based on use of the WHO second international standard[32]. The IgG response to the measles virus was measured by EIA using the measles IgG test system (Zeus Scientific, Branchburg, NJ), and the avidity of measles-specific IgG antibodies was tested by modification of a commercial measles IgG EIA (Captia Measles IgG, Trinity Biotech, Jamestown, NY), as described by Latner et al. and Mercader et al.[32,33].

**Statistics and reproducibility**. For the mitogen-induced intracellular cytokine secretion, CD107a expression assays, phosflow analyses and vaccine antigen-specific Ab responses, the D'Agostino–Pearson omnibus normality test ($\alpha = 0.05$) for normal distribution was performed followed by the Kruskal–Wallis and Dunn's multiple comparison's tests ($\alpha = 0.05$) to identify any potential statistically significant differences in the patterns measured between the three cohorts (Prism version ≥8.0, GraphPad, San Diego, CA). The statistical comparison between the control and cohort-1 subjects depicted in the form of scatter-plot column graphs for the subsets identified through the CITRUS analyses utilized the two-tailed Mann–Whitney $U$ test (Prism version ≥8.0, GraphPad, San Diego, CA). Statistical assessments for the Gag PTE-pool induced intracellular cytokine secretion and CD107a

surface mobilization assays also utilized the D'Agostino–Pearson omnibus normality test ($\alpha = 0.05$) followed by the two-tailed Mann–Whitney $U$ test (Prism version ≥8.0, GraphPad, San Diego, CA). For all statistical analyses, measurements were taken from distinct samples. The figure legends indicate the number times each sample was evaluated.

**Reporting summary**. Further information on research design is available in the Nature Research Reporting Summary linked to this article.

## Results

**HIV+ study cohorts**. Study subjects in cohorts 1 and 2 were born to mothers with a confirmed diagnosis of HIV-1 infection (Supplementary Table 1). The study subjects are part of the HIV/AIDS prevention program operating within the Division of Infectious Diseases at our institution (this program was initiated in 1987). The program is staffed by board-certified infectious disease specialists and operates an on-site laboratory certified to perform viral load testing. All of the study subjects were receiving outpatient care at our institution at the time of enrollment, and none of the subjects were diagnosed with any opportunistic infections at the time of enrollment into the study which might have potentially skewed the lymphocyte responses in any way. The study subjects also received the ART combinations that are part of the accepted clinical standard of care (Supplementary Table 3). Cohort 1 subjects displayed greater consistency in adherence to ART which is reflected in better viral suppression as well as reduced selection of ART resistant virus in cohort 1 versus cohort 2 subjects, after initial viral control and prior to initiation of our lymphocyte function studies (Fig. 1) (Supplementary Table 1).

**Mitogen-induced polyfunctional responses are preserved in both HIV+ cohorts**. The relationship between the degree of virologic control and maintenance or diminution and resurgence of robust T cell polyfunctional responses is less clear cut in pediatric subjects and appears to differ between CD4 and CD8 T cells[4,34–36]. Hence, we wanted to examine cytokine responses in our study cohorts to determine if polyfunctionality was compromised in cohort 2 study subjects. Whole blood samples were stimulated with PMA/Ion to induce cytosolic co-expression of IFNγ, TNFα, IL-2 and IL-21. We were able to measure sixteen different combinations of the four cytokines for each of the indicated lymphocyte subsets (Supplementary Fig. 1). Intriguingly, we detected no evidence of suppressed polyfunctionality in the lymphocyte subsets of the cohort 2 study subjects (Fig. 2). Importantly, the frequencies of CD8+ and CD8− T cell subsets that produced all four cytokines in tandem (that is, they were highly polyfunctional) were the highest in the cohort 2 study subjects (Fig. 2i). Furthermore, for at least three other cytokine combinations, the frequencies of CD8+ or CD8− T cells coproducing these cytokines trended higher in the cohort 2 study subjects (Fig. 2e, h, j–m). Statistically significant differences were only observed for the CD3 negative lymphocytes of cohort 1, where we observed a diminution in the functional response (Fig. 2e, p). When we focused purely on the amplitude of the functional response, in all three cohorts, it was the CD8- T cells producing either IL-2 alone, or IL-2 in combination with TNFα, that displayed the strongest responses (≥~20% of the CD8- T cells displayed these two cytokine expression profiles) (Fig. 2d, g). We also evaluated the degranulation potential of the lymphocytes in all three cohorts by measuring the surface mobilization of CD107a (lysosomal associated membrane protein-1; LAMP-1) in tandem with IFNγ production with and without PMA/Ion treatment

(Supplementary Fig. 2a, b). We detected no evidence of diminished degranulation capacity in the two HIV+ cohorts compared to the healthy control samples. In fact, the CD8+ IFNγ+ T cells in cohort 2 displayed a higher frequency of CD107a surface mobilization compared to the healthy control subjects, while the degranulation potential in the CD8− T cells and CD3- lymphocytes were very comparable among the three cohorts (Supplementary Fig. 2a, b). These data demonstrate that in our perinatally-infected subjects, sub-optimal virological control did not impair mitogen-induced polyfunctionality of the lymphocytes.

**Unsupervised analyses identify unique immune clusters in Cohort 1 study subjects that differ significantly from healthy controls**. We executed unsupervised (automated) clustering analyses (CITRUS) of gated lymphocytes to facilitate the discovery of unique aggregations of lymphocyte populations within our study cohorts (Figs. 3, 4 and Supplementary Fig. 3)[29]. The CITRUS run was repeated a minimum of four times for every group pairing (Control versus Cohort 1, Control versus Cohort 2, and Cohort 1 versus Cohort 2) for assessing the abundance and median cluster characterizations. The only statistically significant differences noted in the CITRUS analyses were between the control samples and cohort 1. When the CITRUS feature being analyzed was the median expression level of the cytokines being measured in the cell populations, we identified six clusters of cells where the level of IFNγ expression in the cohort 1 study subjects was significantly decreased compared to the healthy control samples (Fig. 3 and Supplementary Fig. 3A). Three of these clusters (850028, 849995, and 849955) were comprised of cells that displayed the CD3 negative CD8lo phenotype while the remaining three clusters (850011, 850030, 849960) were made up of cells that were CD3 + CD8lo (Fig. 3). Two of these CD3 + CD8lo clusters (clusters 850011 and 850030), also demonstrated a significant reduction in the median expression of TNFα (Fig. 3). When the CITRUS analyses measured the relative abundance of cell clusters that diverged significantly from the healthy control samples, we identified four clusters that were significantly enriched in the cohort 1 study subjects, and these clusters were comprised of CD3 negative lymphocytes with varying levels of CD8 expression and additional subtle distinctions in IFNγ (clusters 395599 and 395553) as well as IL-2 coexpression (cluster 395564) (Fig. 4 and Supplementary Fig. 3B).

**Functional Gag-specific CD8 T cell responses are detected only in cohort 2 study subjects**. PMA/Ion treatment induces bulk lymphocyte stimulation irrespective of antigenic specificity. Therefore, in order to interrogate HIV-specific T cell responses we stimulated whole blood samples in the presence or absence of the Gag Potential T cell Epitope (PTE) peptide pool (Fig. 5 and Supplementary Fig. 4). We chose Gag peptides based on the observation that Gag epitopes are well conserved in HIV-infected individuals, and Gag-specific T cell responses play an important protective role[17,37–42]. Detectable Gag-specific T cell responses were restricted to the CD8+ T cells and were observed exclusively in Cohort 2 (five out six study cohort 2 subjects had measurable responses) (Fig. 5 and Supplementary Fig. 4g–l, second column). These responses were further restricted to the production of IFNγ and/or TNFα (IL-2 and IL-21 production was not observed) and the magnitude of the IFNγ and/or TNFα response correlated well with frequency of Gag-specific CD8+ T cells displaying surface mobilization of CD107a (LAMP-1) (Fig. 5 and Supplementary Fig. 4g–l, fourth column). Our Gag-specific CD8 T cell data are concordant with our PMA/Ion data which did not reveal any attenuation in the lymphocyte function capacity in cohort 2, as well as with a recent report describing

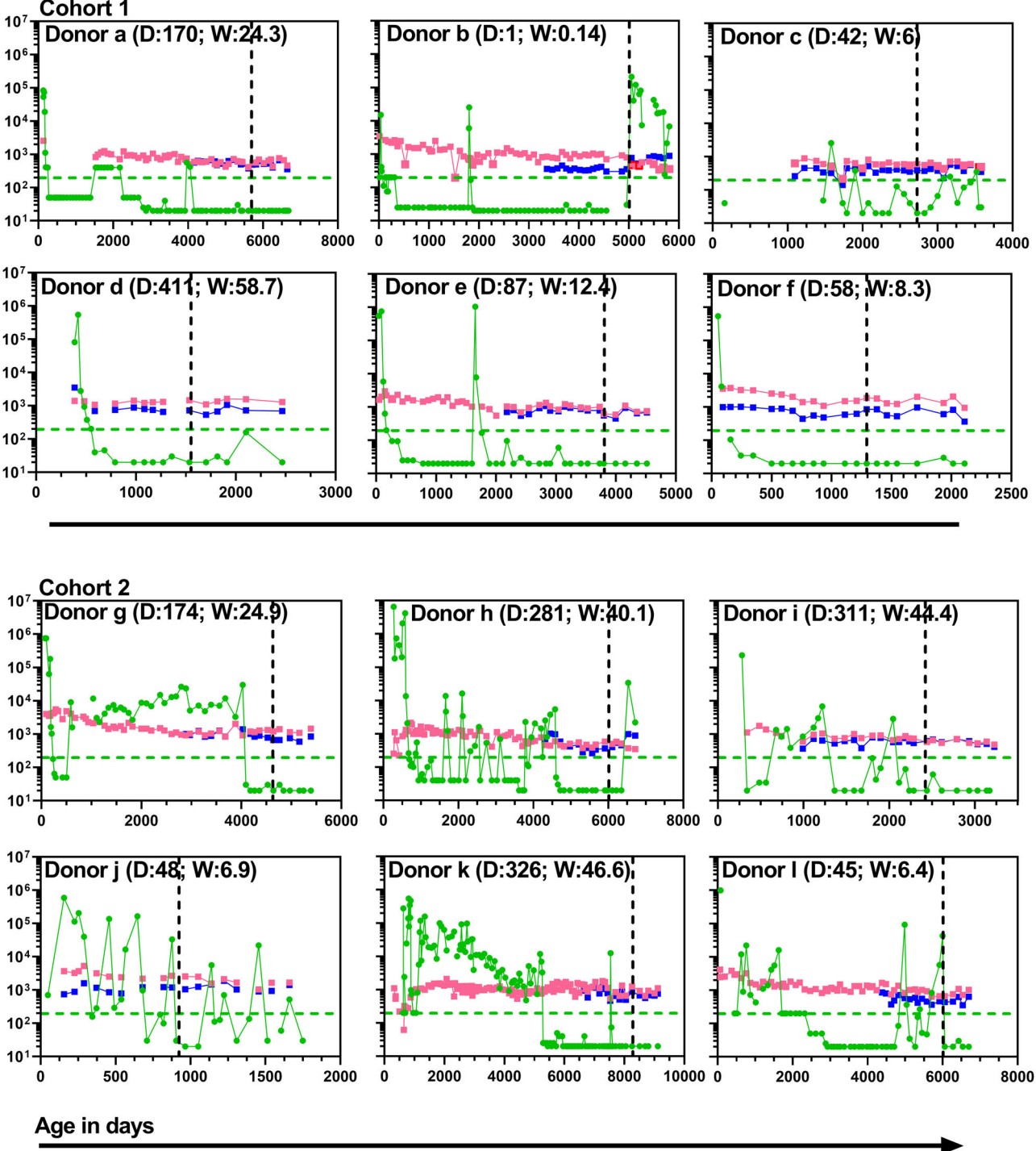

**Fig. 1 Longitudinal assessments of viral load and circulating T cell counts.** For this study the lower limit of detection (LOD) for the plasma viral load was set at 200 copies/mL and is indicated in each plot by the green dashed horizontal line. In some plots the gaps between individual data points are provided where either the viral load or T cell counts were not clinically ascertained in tandem for that particular time point. The numerical values in parentheses in each plot represent the day (D) and week (W) of life when ART was initiated. Green circles: HIV RNA copies/ml of plasma; Pink squares: Number of CD4 T cells/μl of blood; Blue squares: Number of CD8 T cells/μl of blood; Black vertical dashed line: day of life when the immune assessments reported in this study were performed. Cohort 1 (n = 6 biologically independent samples), Cohort 2 (n = 6 biologically independent samples).

viremic non-suppressors that initiated ART at a median age of 24 months of life[4]. The failure to detect Gag-specific T cell responses in cohort 1 are also consistent with numerous reports that demonstrate a diminution or quiescence of the functional T cell response in the face of prolonged viral suppression

following early ART initiation[7,8,43–55]. Additionally, there is no difference in the number of subjects between the two cohorts receiving protease-inhibitors (PI) or non-nucleoside reverse transcriptase inhibitors (NNRTI) (Supplementary Table 3). Hence, it is quite unlikely that the differences in the class of anti-

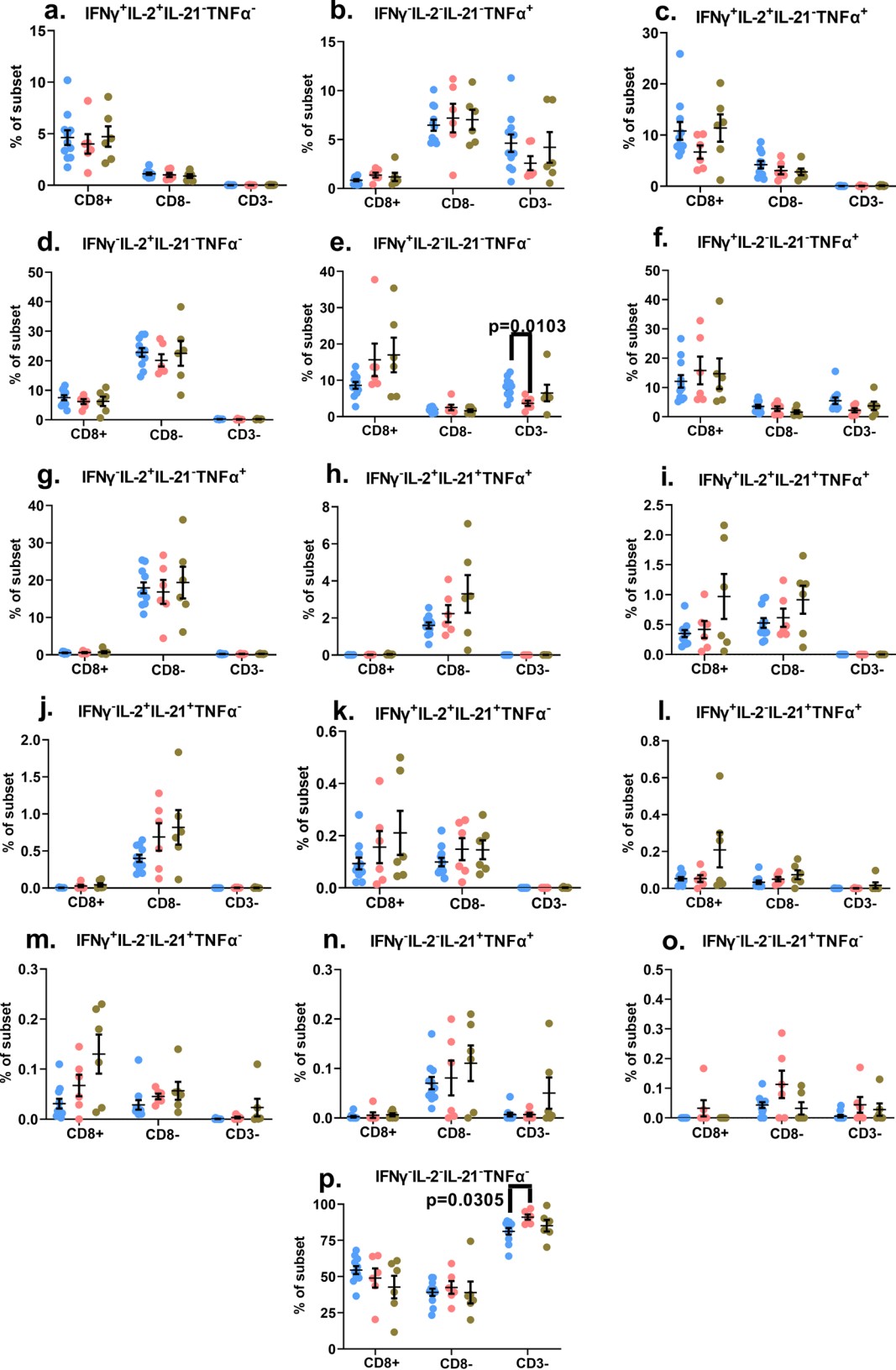

**Fig. 2 Examination of the polyfunctional potential of mitogen-stimulated lymphocyte subsets.** The scatter plot graphs represent background-subtracted frequencies of the indicated lymphocyte subsets producing the specific combination of the four cytokines (listed atop each graph) following PMA/Ionomycin stimulation (**a**–**p**). The data depict the values of each individual study subject as well as the mean ± standard error of mean (SEM) for each group. *p*-values < 0.05 are considered statistically significant. Blue circles: Control subjects; Pink circles: Cohort 1 subjects; Green circles: Cohort 2 subjects. Cohort 1 (*n* = 6 biologically independent samples), Cohort 2 (*n* = 6 biologically independent samples), Controls (*n* = 11 biologically independent samples). Each study subject was evaluated once.

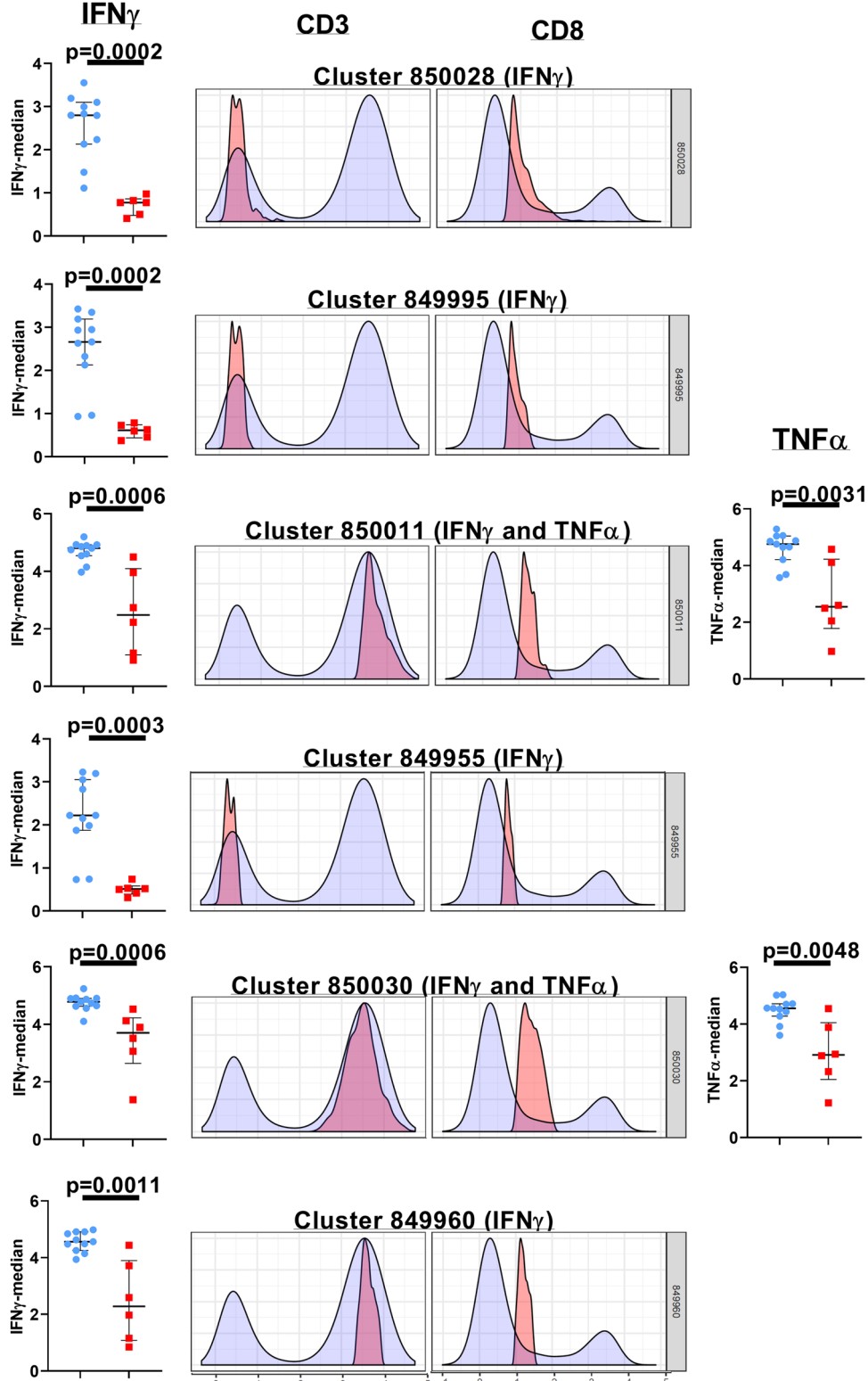

**Fig. 3 Phenotypic resolution and quantitative assessment of functionally suppressed lymphocyte cell clusters in Cohort 1.** The histogram overlay plots for each of the listed cellular clusters depict the CD3 and CD8 expression profiles in that specific cluster (pink) compared to the background population (blue). The scatter-plot graphs shown adjacent to the histogram overlay plots depict the medians with the interquartile ranges of IFNγ (left) and TNFα (right) expression of the lymphocytes within the cluster. In the scatter-plot graphs, the control subjects are indicated by the blue circles, and the cohort 1 subjects are indicated by the red squares. These data are representative of four independent CITRUS runs examining the median expression of the cytokines. A *p*-value < 0.05 is considered significant. Cohort 1 (*n* = 6 biologically independent samples), Controls (*n* = 11 biologically independent samples).

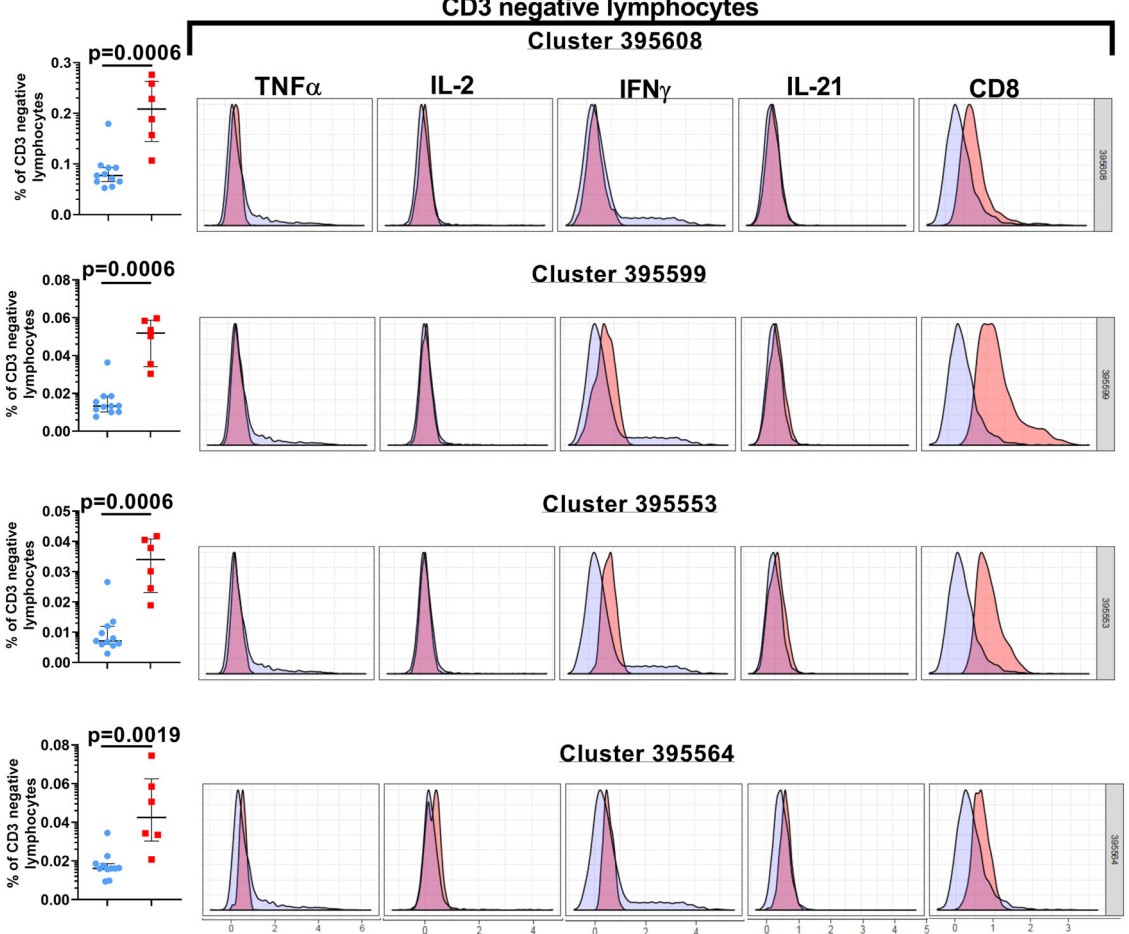

**Fig. 4 Significant enrichment of CD3 negative lymphocyte cell clusters in Cohort 1.** The histogram overlay plots for each of the listed clusters, based on the abundance criterion, depict the phenotype and cytokine expression properties of the cells within that specific cluster (pink) compared to the background population (blue). The scatter plot graphs shown to the left of the histogram overlay plots display the median frequencies and interquartile ranges of the CD3 negative lymphocytes within each cluster. In the scatter-plot graphs, the control subjects are indicated by the blue circles, and the cohort 1 subjects are indicated by the red squares. These data are representative of four independent CITRUS runs examining the abundance of the cytokines. A *p*-value < 0.05 is considered significant. Cohort 1 (*n* = 6 biologically independent samples), Controls (*n* = 11 biologically independent samples).

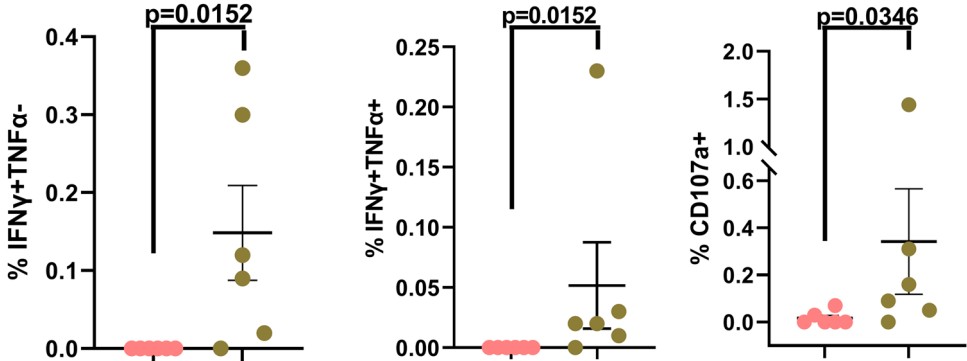

**Fig. 5 Sub-optimal virologic control does not impair HIV-specific CD8 T cell responses.** The scatter plot graphs summarize the % of background-subtracted Gag-PTE-pool specific CD8 T cell responses in cohort-1 and cohort-2 study subjects. Data from individual subjects are displayed in addition to the mean ± SEM of the parameters listed along the *y*-axis. Pink circles: Cohort 1 subjects; Green circles: Cohort 2 subjects. *p*-values < 0.05 are considered statistically significant. Cohort 1 (*n* = 6 biologically independent samples), Cohort 2 (*n* = 6 biologically independent samples). Each study subject was evaluated once.

viral drugs used in the two cohorts is contributing to the differences in the Gag-specific CD8 T cell responses observed in this study, in contrast to what has been reported in some small observational studies[54,56,57].

**Sub-optimal viral control does not subvert lymphocyte signaling.** Previous reports have indicated that signaling pathways involving signaling transducer and activator of transcription 5 (STAT5) and Type 1 IFN are likely affected in HIV+ individuals displaying varying degrees of virologic control[58–65]. Additionally, long-term ART has also been reported to be associated with generalized inflammation-induced serious non-AIDS events (sNAEs)[66–69]. Therefore, we undertook a comprehensive and systematic analyses of the signaling pathways for several immunomodulatory cytokines that regulate T cell function, survival and homeostasis, and we also examined signaling induced through CD3 stimulation (Fig. 6 and Supplementary Figs. 5–7). Whole blood samples were treated with the indicated signaling inputs and the phosphorylation status of the appropriate downstream proximal signaling nodes was measured by phosflow. This approach also offered a quick and sensitive assessment of any underlying hyper-inflammatory state which can be detected by the phosphoprotein signal measured in the absence of any exogenous signaling input (Fig. 6, PBS-treated samples). Our data revealed no signaling impairments in any of the three lymphocyte subsets of the cohort-2 subjects following treatment with the cytokines and anti-CD3 MAb (Fig. 6 and Supplementary Figs. 5–7). In fact, treatment of whole blood samples with IL-2 and IL-15 led to a significant upregulation of the CD8+ T cell associated pSTAT5 signal, and IL-4 treatment significantly enhanced the pSTAT6 signal in the CD8- T cell compartment in cohort 2 study subjects (Fig. 6d, h). When we measured raw median fluorescence intensity (MFI) values of the PBS treated samples, there was evidence of significant basal hyper-phosphorylation in some of the signaling nodes of both HIV+ cohorts. Specifically, STAT1 basal hyper-phosphorylation was observed in the CD8+ T cells of the cohort 1 samples, as well as CD3− lymphocytes of both cohort 1 and 2 study subjects (Fig. 6a, c). STAT5 basal hyper-phosphorylation was observed exclusively in cohort 2 study subjects (Fig. 6d–f), while STAT6 and STAT3 basal hyper-phosphorylation was detected only in cohort 1 samples (Fig. 6g–i, l). Calculating the fold change in the phosphorylation signal (MFI of the signaling node following cytokine treatment/MFI of the signaling node following PBS treatment), demonstrated that the only attenuation detected was in the Cohort 1 samples relative to the signal measured in healthy control samples. Specifically, IL-4 treatment-induced STAT6 phosphorylation was significantly reduced in CD8+ T cells, while the IL-21 induced pSTAT3 signal was reduced in CD8− T cells as well as CD3− lymphocytes (Supplementary Figs. 5c, 6d, 7c). Additionally, CD3 stimulation-induced ZAP70 phosphorylation was significantly dampened in both CD8+ and CD8− T cells in the cohort 1 study subjects (Fig. 6p, q; Supplementary Figs. 5f, 6f). Overall, we observed no blunting of the signaling potential of pediatric subjects with relatively impaired virologic control (cohort 2). On the other hand, our findings do suggest an altered CD3-engagement associated activation threshold, and dampened IL-4 and IL-21 signaling in cohort 1 subjects. A basal hyper-activated signaling state affecting multiple cytokine signaling pathways was also evident to a greater degree in HIV+ subjects that displayed better virologic control (six assessments in cohort 1 and four in cohort 2 displayed basal hyper-phosphorylation). These data suggest the persistence of low-grade inflammation in both HIV+ cohorts, superimposed with relative dulling of the signaling potential in circulating lymphocytes in cohort 1. These findings are also consistent with our intra-cellular cytokine data which indicate a relative diminution observed in cohort 1 subjects.

**Childhood vaccines-induced serum Ab responses are similar in healthy controls and HIV+ study subjects.** To complement our analyses of cell-mediated lymphocyte responses, we also examined humoral immune responses to childhood vaccines in our study subjects (Fig. 7). IgG responses to the Varicella Zoster Virus (VZV), tetanus toxoid (TT), Haemophilus influenzae b (Hib) were measured by enzyme immunoassays (EIA)[70,71]. In addition, we also assessed antibody responses to the measles virus using both an EIA as well as the gold standard method for determining measles immunity, an in vitro plaque-reduction neutralization assay[32]. We detected antibody responses against the vaccine targets in all three cohorts although in both HIV+ cohorts the amplitude of these responses was lower than that observed in the healthy control samples (Fig. 7 and Supplementary Fig. 8). High measles IgG-avidity test results were demonstrated in all subjects with sufficient IgG to test indicating that all three groups were capable of functional avidity maturation. We also plotted the data that measured the duration between the latest vaccine dose for each study subject and the time point at which the serum sample was obtained for these analyses (Fig. 7 and Supplementary Fig. 8). This latter assessment suggested that the perceived differences in the amplitude of the Ab responses between the three cohorts likely reflected the temporal disparity in the duration between the latest vaccine dose and the timing of this assessment rather than any intrinsic defect in the humoral immune responses elicited by vaccination in our HIV+ study subjects. These results demonstrate that impaired virologic control did not significantly erode vaccine-induced protective immunity, and they also indirectly argue against the existence of a functional deficit in the germinal center reactions mediated by T follicular helper (Tfh) cell responses that contribute to heterologous humoral immunity, in the two HIV-infected cohorts.

**Discussion**
Delineating the correlates of immune-mediated control and defining the parameters of immune-dysregulation in early-treated HIV+ children are key elements of the concept to attempt a potential cure in this population. In this study, we demonstrate that lymphocyte functional fitness (as evaluated in this study) in perinatally infected HIV+ pediatric subjects with early ART initiation and sub-optimal virologic control (cohort 2), was essentially preserved, and better aligned with the responses measured in HIV-negative healthy control donors compared to cohort 1 subjects that displayed better virologic control. PMA/Ion-induced polyfunctional T cell responses measured in both HIV+ cohorts were similar to healthy controls, and the only evidence of significant suppression of cytokine responses were noted in the CD3 negative lymphocytes of cohort 1 (Fig. 2). When these PMA/Ion-induced cytokine production data were further queried using CITRUS, the results corroborated the absence of any immune function aberrations arising as a result of impaired virological control (cohort 2). These analyses also uncovered distinctive cellular subsets that segregated with the CD3 negative lymphocytes that were significantly over-represented in cohort 1 (Fig. 4). These CD3 negative lymphocyte subsets co-expressed CD8 variably above background levels, with blunted cytokine responses (Fig. 4). These functional impairments were further confirmed when we separately ran the CITRUS analyses focusing on the median expression of the cytokines (Fig. 3). This latter assessment also identified two unique clusters of CD3+ CD8lo T cells that were deficient in both IFNγ and TNFα production (Fig. 3). Collectively, the outcomes of these unsupervised analyses supplemented the findings of the conventional (supervised) analytical approaches which had initially identified two groups of the Boolean gating cytokine combinations in cohort 1 donors that varied significantly from

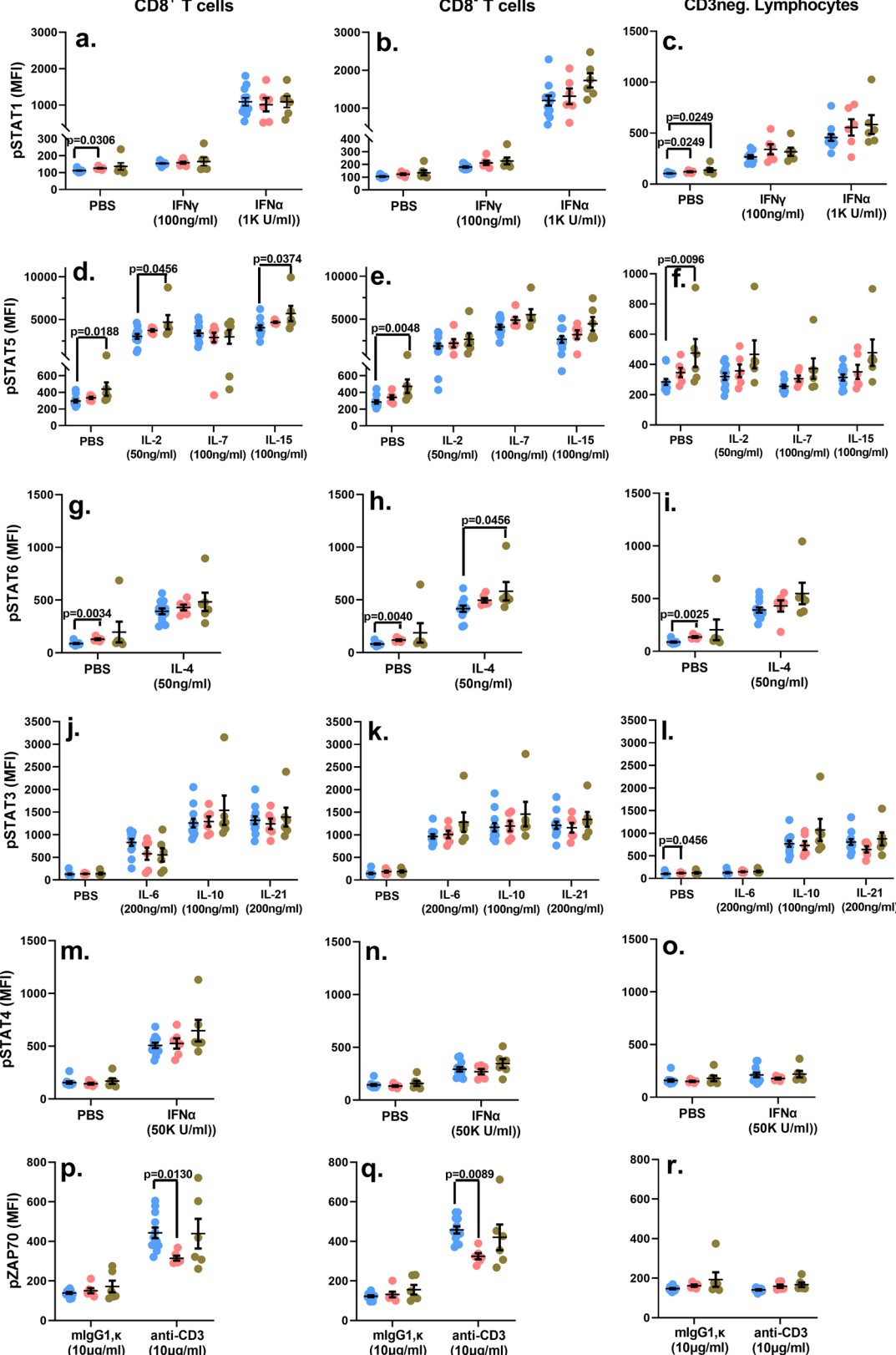

**Fig. 6 Raw MFI values of the indicated phosphoproteins.** Whole blood samples were treated with the listed signaling inputs (x-axes) and the MFI values for the phosphorylation signal of the corresponding signaling nodes (y-axes) are depicted for each individual study subject in addition to the mean ± standard error of mean (SEM) for each group (**a–r**). Blue circles: Control group; Pink circles: Cohort 1 group; Green circles: Cohort 2 group. p-values < 0.05 are considered statistically significant. Each study subject was evaluated once. Cohort 1 (n = 6 biologically independent samples), Cohort 2 (n = 6 biologically independent samples), Controls (n = 11 biologically independent samples).

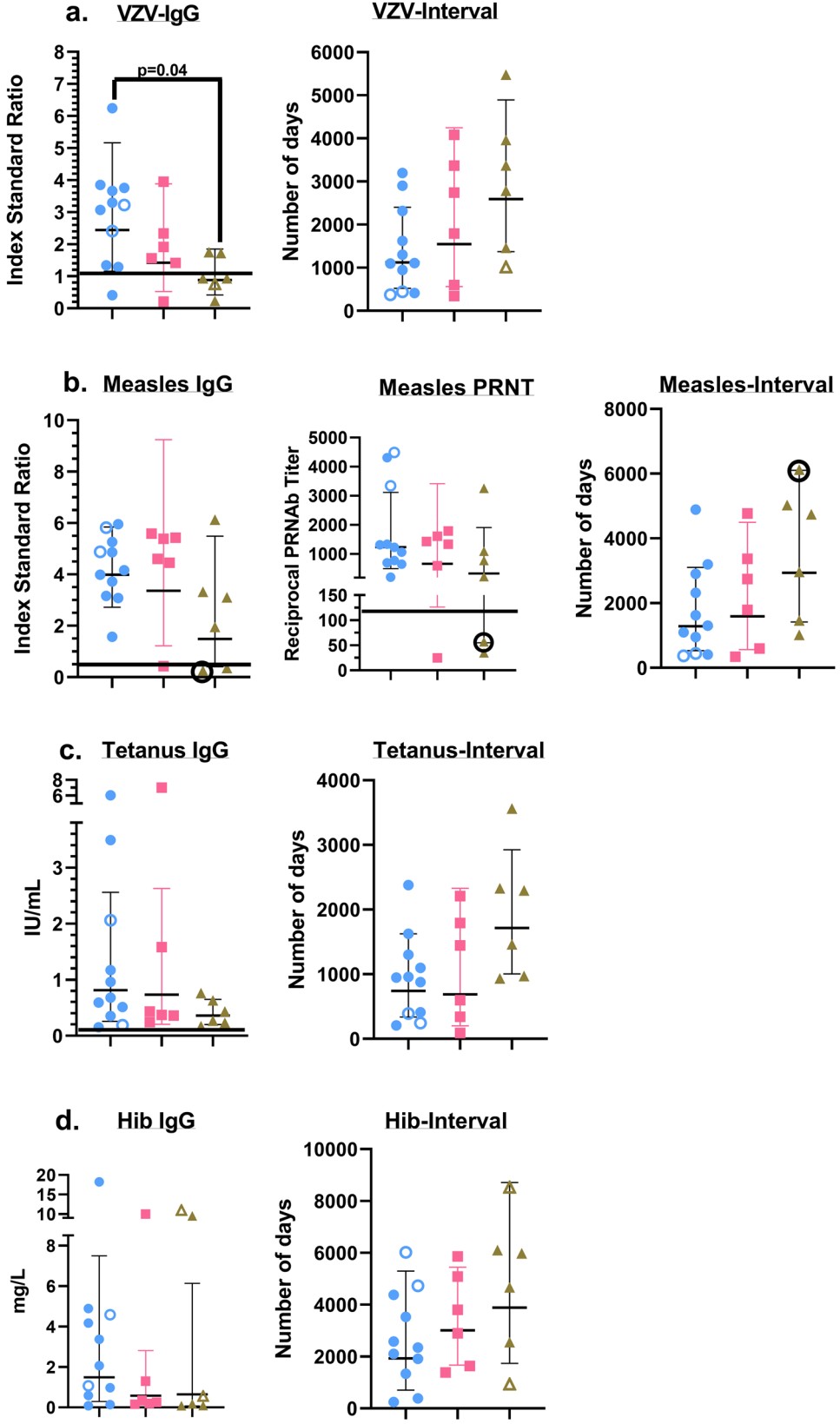

the healthy control donors (specifically, the diminution of CD3 negative lymphocytes that expressed the IFNγ alone, and the enhancement of the CD3− lymphocyte group that did not generate any of the four tested cytokine responses) (Fig. 2e, p).

The interrogation of virus-specific T cell responses revealed that functional Gag-specific CD8 T cell responses were detected only in

cohort 2. We selected Gag as the HIV antigen of choice because it is an important CD8 T cell target and is highly conserved[17,37–42]. Five out of the six cohort 2 study subjects displayed detectable functional Gag-specific T cell responses after stimulation with the Gag PTE peptide pool (Fig. 5 and Supplementary Fig. 4). The peptides in this pool are 15 amino-acids (aa) in length and harbor

**Fig. 7 Serum Ab responses to childhood vaccines.** IgG responses to the listed childhood vaccines are represented as geometric means ± geometric SD along with values for each individual study subject. The horizontal line above the x-axes, wherever depicted in the Ab plots, indicates the kit manufacturer recommended cut-off for a positive response (VZV and Measles IgG), or the currently accepted optimal protective Ab level or titer (Tetanus IgG, Hib IgG and Measles PRN titer)[32,70,71]. The plots that display the number of days along the y-axis, depict the time interval (as geometric means ± geometric SD along with values for each individual study subject) between the latest vaccine dose and the serological assessment performed for this study. Blue circles: Control group; Pink squares: Cohort 1 group; Green triangles: Cohort 2 group. The unfilled individual symbols denote study subjects that received fewer vaccine doses compared to their peers, due to their age at the time the serum sample was obtained for the study: **a** VZV IgG: Filled symbols (2 doses of the vaccine); unfilled symbols (1 dose of the vaccine). **b** Measles serology: Filled symbols (2 doses of the vaccine); Filled symbols, circled (3 doses of the vaccine); unfilled symbols (1 dose of the vaccine). **c** Tetanus IgG: Filled symbols (5–7 doses of the vaccine); unfilled symbols (4 doses of the vaccine). **d** Hib IgG: Filled symbols (4–5 doses of the vaccine); unfilled symbols (3 doses of the vaccine). A p-value < 0.05 is considered significant. Each study subject was evaluated once. Cohort 1 (n = 6 biologically independent samples), Cohort 2 (n = 6 biologically independent samples), Controls (n = 11 biologically independent samples).

9 aa long sequences that are potential T cell epitopes, but they are not defined peptide epitopes matched to our study subject individual haplotypes. Hence, it is possible that peptide-HLA mismatch could be one reason why we could not detect a CD8 T cell response in donor l (Supplementary Fig. 4, second and fourth columns) in cohort 2, and Gag-specific CD4 T cell responses in either cohort. Alternatively, the Gag-specific CD8 T cell response might be present in this donor but below the limit of detection of our assay. Furthermore, the failure to clearly detect cytokine-producing Gag-specific CD4 T cell responses is concordant with what has been previously reported in perinatally-infected subjects that initiated ART between 3 and 12 months of life[72]. It's theoretically also possible that even though functional Gag-specific circulating T cell responses were undetectable, circulating T cell responses to non-Gag HIV antigens might still be present, and HIV-specific (including Gag-specific) T cell responses might be detectable in secondary lymphoid and tertiary non-lymphoid tissues in cohort 1 study subjects. However, it has also been reported in pediatric slow progressors (that have clearly detectable viremia) that lower viral loads are accompanied by a dominant Gag/Pol-specific T cell response, and conversely, higher viral loads are marked by a greater breadth as well as magnitude of non-Gag/Pol-specific T cell responses[7]. Cohort 2 Gag-specific CD8 T cell responses were restricted to IFNγ and TNFα production and we did not observe any IL-2 or IL-21 production by these cells. This finding is also consistent with descriptions of Gag-specific CD8 T cell responses in perinatally infected subjects that initiated ART between 3 and 12 months of life, and we do not believe that our inability to detect these additional cytokines was affected by our assay protocol in any way, as we have recently published using the identical stimulation protocol with defined EBV peptide epitopes to induce IFNγ, TNFα, and IL-2 production by EBV-specific CD8 T cells[26,72]. Several studies have examined T cell responses in HIV+ pediatric subjects, and overall our data are concordant with an overwhelming majority of these reports that have demonstrated that early ART initiation and sustained virologic suppression in HIV-infected children can attenuate the virus-specific T cell response (which is clearly induced and detectable at birth in most infected infants), suggesting a link between measurable viremia and maintenance of a detectable virus-specific immune response[7,8,43–55,73,74].

The immune response is regulated by carefully orchestrated signaling networks that guide the expansion and contraction of the response, optimal effector function and establishment of immune memory. Consequently, disruption of these signaling pathways can lead to a systemic unraveling of immune homeostasis. Relatively few studies have examined the contribution of dysregulated immune signaling pathways to immune dysfunction in the setting of HIV infection, and to the best of our knowledge an overwhelming majority of these studies have been performed in adults[58–63,65]. Dampened IL-2 induced phosphorylation of STAT5,

and T cell receptor (TCR) cross-linking mediated phosphorylation of ZAP70 have been reported in T cells of untreated HIV+ adults that display progressive disease, while long-term non-progressors do not display these impairments[58,59,63]. Additionally, systems immunology approaches examining T cells in HIV-infected, adult elite controllers have identified the STAT5 pathway as a component of the molecular signatures associated with memory T cell survival in these subjects[64]. It has also been reported that IL-7-induced STAT5 phosphorylation can be inhibited in viremic adult patients[60,63]. Our analyses that included stimulation with IL-2 as well as IL-7, which is a key mediator of T cell homeostasis, did not reveal any signaling-associated deficits in T cell-associated STAT5 phosphorylation in either HIV+ cohort (Fig. 6d–f; Supplementary Figs. 5b, 6b). Furthermore, treatment with IL-15, which plays a prominent role in NK cell homeostasis, similarly did not highlight any deficits in STAT5 phosphorylation[75] (Fig. 6d–f; Supplementary Figs. 5b, 6b, 7e). Published evidence also indicates that HIV proteins such as Vpu and Nef can actively block IFN-induced STAT1 phosphorylation, while Vif can degrade STAT1[61]. Once again, we did not observe any perturbations in IFNγ-mediated phosphorylation of STAT1, and Type 1 IFN-mediated phosphorylation of STAT1 or STAT4 in the CD8+ or CD8− T cells as well as CD3-negative lymphocyte in our three study cohorts (Fig. 6a–c, m–o; Supplementary Figs. 5a, e; 6a, e; 7a, f). However, signaling responses, induced following IL-4 and IL-21 treatment, were significantly attenuated in cohort 1 subjects (Supplementary Figs. 5c, 6d, 7c). One of the intriguing findings in our report is that all of the significant signaling impairments, measured as the fold change in phosphorylation status, were noted in study subjects displaying better virological control (cohort 1) (Supplementary Figs. 5–7). This pattern is in contrast to those observed in adult subjects, and they reinforce the notion that HIV infection affects the immune response differently in children versus adults[58–63,65]. Considering the results of the mitogen and Gag-induced cytokine production, coupled with identification of functionally suppressed lymphocyte clusters in cohort 1 samples using CITRUS, the signaling data are concordant with the notion of relatively dampened functionality in the circulating lymphocyte pool in cohort 1 subjects. We also observed that signaling through the TCR and subsequent phosphorylation of ZAP70, was attenuated in both the CD8+ and CD8− T cells in the cohort 1 study subjects (Fig. 6p, q; Supplementary Figs. 5f, 6f). Given the lack of a functional Gag-specific T cell response in cohort 1, the depressed pZAP70 signal data in cohort 1 is also concordant with the notion of T cell functional quiescence associated with sustained virologic control following early ART initiation in pediatric subjects. We have recently published that anti-CD3 activation-induced phosphorylation of ZAP70 is similar between naive and memory T cells, hence differences in the relative proportions of naive and memory T cell subsets between cohort 1 and 2 would be an unlikely contributor to the blunted ZAP70 phosphorylation observed in our cohort 1 subjects[26].

Furthermore, the median age of cohort 1 subjects is greater than that of the control subjects (Supplementary Table 1), hence it is very likely that the cohort 1 subjects harbor a greater proportion of memory T cells than the control subjects. This scenario further argues against the possibility that harboring a greater proportion of naïve T cells could be a contributor to the dampened signaling observed in cohort 1 subjects. Additionally, it is unlikely that the attenuated responses in cohort 1 are reflective of an increase in the proportion of exhausted cells since enhanced expression of conventional markers of exhaustion, such as PD-1, correlate better with the magnitude of the HIV-specific response in children, hence are more likely to be associated with cohort 2[10]. Basal hyperphosphorylation is yet another feature of aberrant signaling that has been reported to co-segregate with sub-optimal virological control in adults with HIV[58]. Persistent systemic inflammation and low-grade immune activation have also been described to contribute to premature immune aging ("inflamm-aging")-associated serious non-AIDS events (sNAE) in HIV+ patients on long-term ART[66–69]. Our data demonstrating basal hyperphosphorylation of select signaling nodes (STAT1 in both cohorts, STAT6 and STAT3 in cohort 1, and STAT5 in cohort 2) suggest that immune hyper-activation is also evident in pediatric subjects that initiate ART early in life, and exert optimal viral control, and that these individuals might also potentially suffer from the negative consequences of "inflamm-aging" in due course similar to adult HIV+ subjects on long-term ART.

Prevailing data suggest intriguing differences in the composition of the HIV reservoir between adults and children[7,18]. Studies have also highlighted differences in the size of the HIV reservoir in perinatally infected children that display differences in the timing of ART initiation after HIV diagnosis and the timing of acquisition of Epstein Barr virus (EBV) and cytomegalovirus (CMV), as well as the subsequent pattern of HIV, EBV and CMV suppression[76,77]. The recent description of immunophenotypic markers that can be utilized to estimate the size of the HIV reservoir in perinatally infected, ART-treated subjects is also noteworthy[78]. Our future efforts will build on these emerging lines of scientific evidence and we will endeavor to comprehensively characterize the HIV reservoir in our study subjects.

In summary, our results demonstrate that sub-optimal viral control (to the extent displayed by our cohort 2 pediatric study subjects) did not compromise the functional fitness of the immune parameters we examined in this study, highlighting the resilience of the immune response within this cohort. As mentioned previously, the viral reservoir in HIV+ children is not heavily populated with escape variants and it has also been reported that the CD8 T cell responses in children do not appear to be severely constrained by original antigenic sin as they are better able to mount variant-specific CD8 T cell responses compared to adults[7–9,17,18]. With respect to HIV+ children on ART with superior viral control (cohort 1), our findings suggest that periodic administration of a vaccine engineered to selectively stimulate HIV (and low-fitness variants)-specific CD8 T cells, could be considered to reverse the functional quiescence. We further speculate that such an approach could potentially contribute to immune-mediated "block and lock" strategies (wherein immune selection pressure forces the virus to establish latency in transcriptionally-silent regions of the genome) that might also eventually maintain viral suppression without ART, similar to what is observed in aviremic, adult elite-controllers[79]. Collectively, these results advance our understanding of pediatric HIV infection associated host pathogen dynamics and provide impetus to further explore immunological approaches to attempt a cure in perinatally infected HIV+ pediatric patients. One of the limitations of this study is the small number of study subjects, which is reflective of the relative rarity of perinatally acquired HIV infection in the US[5]. Therefore, our future efforts will involve expanding the pool of potential study subjects to determine if the findings from this report are more broadly applicable in the setting of perinatally acquired HIV infection with early ART initiation and divergent patterns of virological control.

## Data availability
All relevant data including the source data are within the paper and its supporting (supplemental) information files.

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

## Acknowledgements

This study was generously funded by a Developmental Core Pilot Award from the Third Coast Center for AIDS Research, a National Institutes of Health (NIH, Bethesda, MD) funded center (P30-AI117943). We would like to express our sincere gratitude to our study subjects and their parents/legal guardians for agreeing to participate in this study. We would also like to thank Dr. Maheen Quadri, MD, in the Academic General Pediatrics clinic at our institution for help with recruiting healthy control subjects for this study, and Jannie Stewart, Senior Clinical Laboratory Assistant (Infectious Diseases) for outstanding phlebotomy service support. We are also grateful to Ian Taylor (BD-Biosciences) and Geoff Kraker, Qianjun Zhang, Qihao Qi, Yuko Nakane (Beckman Coulter) for helpful discussions, guidance and tutorials relating to tSNE and CITRUS analyses. The findings and conclusions in this report are those of the authors and do not necessarily represent the official position of the Centers for Disease Control and Prevention.

## Author contributions

A.K. designed the study, secured the funding, performed experiments, analyzed the data, wrote and edited the manuscript. W.J.M. helped design the study and reviewed the manuscript. B.M.S., J.C. and G.L. performed experiments and reviewed the manuscript. R.T.D. assisted with securing funding for the study. R.W. assisted with specimen acquisition and reviewed the manuscript. S.B.S., K.M., S.M. and C.J.H. performed experiments and reviewed the manuscript. The National Institutes of Health (NIH, Bethesda, MD) funded this study but played no role in the study design, data analyses or manuscript preparation.

## Competing interests

W.J.M. has received grants or contracts, unrelated to this study, from the following entities: Ansun BioPharma, Astellas Pharma, AstraZeneca, Janssen Pharmaceuticals, Karius, Merck, Genentech, Gilead, Melinta Therapeutics, Nabriva, Seqirus and Tetraphase Pharmaceuticals. The rest of the authors have no competing interests to declare.
