## [Peer Review File · Communications Medicine]

Reviewers' comments:

Reviewer #1 (Remarks to the Author):

The authors have analyzed T and B cell mediated immunity in a cohort of HIV-1 infected pediatric patients with varying degrees of viral control and adherence to antiretroviral therapy. The main finding is that polyfunctional T cells, including HIV-specific T cells, seem to be stronger in patients with more limited adherence to ART and incomplete suppression of plasma viremia. Overall, this is technically-well conducted study. Specific comments:

1. In authors describe three study cohorts, a table should be included into the manuscript that describes the clinical and demographical characteristics of the study patients in more detail.
2. For participants in group 1 and 2, details should be provided regarding time of HIV diagnosis, time of ART initiation, ART regimens, clinical outcomes etc. Evolution of drug resistance (specifically in cohort 2) would also be of interest.
3. In Figure 3 and 4, a computational analysis of T cell responses is presented, and comparisons between cohort 1 and the control group (HIV-negative persons) are shown. Why are data from cohort 2 not included?
4. Figure 5 describes HIV-specific T cells responses; flow cytometry dot plots are shown. A high-level analysis of this data including statistical comparisons between group 1 and group 2 should be presented.
5. In figure 7, the authors show single data plots, which is preferable. Data in Figure 2 and 6 should be shown in a similar way as in Figure 7.
6. HIV reservoir cells were not analyzed in this report, but such studies would be of interest. The authors should be consider viral reservoir evaluations in future studies and discuss such plans in more detail in this report.

Reviewer #2 (Remarks to the Author):

Khanolkar a. et al studied the T cells functionality in perinatally HIV-infected (PHIV) children starting ART early after life and demonstrated that lack of virological control does not affect T cells functionality. The work could be of relevance to the field of pediatric HIV.

Major concerns come from the very small sample size, which makes the results poorly solid. Furthermore, the authors do not take into consideration the importance of the timing of ART start in PHIV, which is otherwise paramount. I have some several concerns that I listed below.

1- Time of ART start. There are numerous evidence supporting that the time of ART start in children is directly associated with preservation of immune function(eg. Rinaldi S. et al *J Immunol.* 2020;204(3):540-549. doi:10.4049/jimmunol.1900856 ; <https://doi.org/10.3389/fimmu.2021.662894> ; <https://doi.org/10.1155/2012/805151>). This aspect about timing of ART start is totally ignored by the authors and warrant discussion. Stating that all children initiated ART within the 1st year of life is not sufficient to support homogeneity of the cohort, thus data on time of ART start and its possible association with immune function needs to be shown, analysed and discussed.

2- Related to point 1 – time on ART needs to be analysed as well in the contest of immune reconstitution: does time of ART exposure affect the immune functionality?

3- Gag-specific CD8 T cells responses. As regards to this dataset, authors are failing to provide all possible scenarios supporting this result. As demonstrated by others 'The generation of an antigen-specific memory cell pool is a dynamic process that evolves from naïve cells to a memory cell pool able to rapidly react upon antigenic re-stimulation. In perinatally HIV-infected children treated with ART early after birth with sustained long-term viral suppression, this process is influenced by both the limited duration of antigen exposure after birth, and the distinct neonatal immune system (Klein N. et al. Lancet Infect Dis 2015,15:1108–1114; Cotton MF et al. Lancet 2013,382:1555–1563). Gp140-specific immune responses in such ET children appears to be low or absent (Luzuriaga K. J Virol 2000,74:6984–6) (taken from: AIDS 2020 Apr 1;34(5):669-680. doi: 10.1097/QAD.0000000000002485). This aspect warrant discussion and the time of ART start and exposure are fundamental parameters to be taken into account before drawing misleading conclusions. As side note, in fact, group II which is the one starting ART later also displays a better anti-HIV response due probably to prolonged antigen exposure.

4- Childhood vaccines-induced serum Ab responses. Authors need to clearly indicate timing of last received vaccination to validate these results.

5- Cytokine production. It is important here to look at the baseline unstimulated levels in group I, II and HC. Delayed ART-start (group II), can be associated with basal higher levels of immune activation, cytokine production et.. Thus, higher level in this case may indicate a pro-inflammatory state consistent with previous literature, rather than suggest an improved functionality.

6- The authors should comment on the fact that starting ART-later (group II) is also associated with poor virological control (supp table 1), in line with previous literature.

We thank the reviewers for their comments. A point-by-point response to each reviewer comment is provided below. The revisions in the main manuscript file are highlighted in blue.

Reviewers' comments:

Reviewer #1 (Remarks to the Author):

The authors have analyzed T and B cell mediated immunity in a cohort of HIV-1 infected pediatric patients with varying degrees of viral control and adherence to antiretroviral therapy. The main finding is that polyfunctional T cells, including HIV-specific T cells, seem to be stronger in patients with more limited adherence to ART and incomplete suppression of plasma viremia. Overall, this is technically-well conducted study. Specific comments:

1. In authors describe three study cohorts, a table should be included into the manuscript that describes the clinical and demographical characteristics of the study patients in more detail.

AK Response: Please refer to the revised Supplementary Table 1 for the details requested.

2. For participants in group 1 and 2, details should be provided regarding time of HIV diagnosis, time of ART initiation, ART regimens, clinical outcomes etc. Evolution of drug resistance (specifically in cohort 2) would also be of interest.

AK Response: Please refer to the revised Supplementary Tables 1 and 2 for the details requested. We are not entirely sure we understand what the reviewer is referring to in terms of “clinical outcomes”. All of the study subjects were receiving outpatient care at our institution at the time of enrollment, and none of the subjects were diagnosed with any opportunistic infections at the time of enrollment into the study which might have potentially skewed the lymphocyte responses in any way. We have also added this note in the Results section (lines 132-135).

3. In Figure 3 and 4, a computational analysis of T cell responses is presented, and comparisons between cohort 1 and the control group (HIV-negative persons) are shown. Why are data from cohort 2 not included?

AK Response: We had mentioned in lines 489-492 of the original submission that “The CITRUS run was repeated a minimum of four times for every group pairing (Control versus Cohort 1, Control versus Cohort 2, and Cohort 1 versus Cohort 2) for assessing the abundance and median cluster characterizations”. The only statistically significant differences noted in the CITRUS analyses were between the control samples and cohort 1. We have now added this information in the Results section. Please refer to lines 171-174.

4. Figure 5 describes HIV-specific T cells responses; flow cytometry dot plots are shown. A high-level analysis of this data including statistical comparisons between group 1 and group 2 should be presented.

AK Response: Summary data with statistical comparisons depicted as scatter dot plots have now been included in the manuscript. These data are listed as Figure 5 in the manuscript. The original Figure 5 is

now listed as Supplementary Figure 4. A notation relating to the statistical comparison for the new Figure 5 is also provided in the Methods section (lines 525-528).

5. In figure 7, the authors show single data plots, which is preferable. Data in Figure 2 and 6 should be shown in a similar way as in Figure 7.

AK Response: Done as requested.

6. HIV reservoir cells were not analyzed in this report, but such studies would be of interest. The authors should be consider viral reservoir evaluations in future studies and discuss such plans in more detail in this report.

AK Response: Done as requested. Please refer to lines 379-386 in the revised version of the manuscript.

Reviewer #2 (Remarks to the Author):

Khanolkar a. et al studied the T cells functionality in perinatally HIV-infected (PHIV) children starting ART early after life and demonstrated that lack of virological control does not affect T cells functionality. The work could be of relevance to the field of pediatric HIV.

Major concerns come from the very small sample size, which makes the results poorly solid. Furthermore, the authors do not take into consideration the importance of the timing of ART start in PHIV, which is otherwise paramount. I have some several concerns that I listed below.

AK Response: We acknowledge that our sample sizes are small, but this is directly related to the fact that perinatally-acquired HIV in the United States is very rare. As mentioned in lines 62-64 of the original submission (and this revised submission) "...perinatal HIV transmission accounted for only 65 out of the ~38,000 new HIV diagnoses in the US in 2018". The study subjects for cohorts 1 and 2 were carefully selected based on the criteria we had set for discriminating the two groups taking into account the timing of ART initiation after HIV diagnosis, and the subsequent establishment, maintenance or disruption in virologic control. Given that this was a pilot study that was limited to the HIV patients at our institution, it will be extremely difficult at this stage to add additional study subjects that meet the criteria we laid out for the study, since the only two additional subjects that met our study criteria declined to participate in the study when approached to provide informed consent for the study. However, we will endeavor to partner with other institutions for future studies to expand our study cohorts and build on the findings of this report. Moreover, reviewer 1 who has specifically listed HIV in children as an area of his/her expertise, did not raise the issue about sample size in this study perhaps reflecting his/her awareness of how rare perinatally-acquired HIV infection is in the United States.

1- Time of ART start. There are numerous evidence supporting that the time of ART start in children is directly associated with preservation of immune function(eg. Rinaldi S. et al J Immunol. 2020;204(3):540-549. doi:10.4049/jimmunol.1900856 ; <https://doi.org/10.3389/fimmu.2021.662894> ;

<https://doi.org/10.1155/2012/805151>). This aspect about timing of ART start is totally ignored by the authors and warrant discussion. Stating that all children initiated ART within the 1st year of life is not sufficient to support homogeneity of the cohort, thus data on time of ART start and its possible association with immune function needs to be shown, analysed and discussed.

AK Response: Supplementary Table 1 in the original submission did indicate the median time in terms of the “week-of-life” when ART was initiated for both cohorts, and this difference in timing of ART initiation between the two cohorts is **not** statistically significant ($p=0.3939$, Mann Whitney test). We have now added data to the revised Supplementary Table 1 which provides additional details on the age at which ART was initiated for the study subjects in both cohorts. We have also added information depicting the day (D) and week (W) of life when ART was initiated for each subject in Figure 1. Furthermore, we cited the Rinaldi et al paper and a relevant manuscript from Cotugno et al in this report (References 62 and 63 in the original submission, and now references 66 and 67 in the revised manuscript). The references provided by the reviewer above have also now been cited in the revised manuscript (references 45-47). Additionally, there is no difference in the number of subjects between the two cohorts receiving protease-inhibitors (PI) or non-nucleoside reverse transcriptase inhibitors (NNRTI) (Supplementary Table 2). Hence, it is quite unlikely that the differences in the class of anti-viral drugs used in the two cohorts is contributing to the differences in the Gag-specific CD8 T cell responses observed in this study, in contrast to what has been reported in some small observational studies (PMID: 22550537). This point has also now been added to the text (lines 203-208).

2- Related to point 1 – time on ART needs to be analysed as well in the context of immune reconstitution: does time of ART exposure affect the immune functionality?

AK Response: Overall, time on ART is greater in cohort 1 than cohort 2 owing to more consistent adherence to ART in cohort 1 that led to both the better virologic control and lesser selection for drug resistance. This point has been added to the text (lines 95-97).

3- Gag-specific CD8 T cells responses. As regards to this dataset, authors are failing to provide all possible scenarios supporting this result. As demonstrated by others ‘The generation of an antigen-specific memory cell pool is a dynamic process that evolves from naïve cells to a memory cell pool able to rapidly react upon antigenic re-stimulation. In perinatally HIV-infected children treated with ART early after birth with sustained long-term viral suppression, this process is influenced by both the limited duration of antigen exposure after birth, and the distinct neonatal immune system (Klein N. et al. Lancet Infect Dis 2015,15:1108–1114; Cotton MF et al. Lancet 2013,382:1555–1563). Gp140-specific immune responses in such ET children appears to be low or absent (Luzuriaga K. J Virol 2000,74:6984–6) (taken from: AIDS 2020 Apr 1;34(5):669-680. doi: 10.1097/QAD.0000000000002485). This aspect warrant discussion and the time of ART start and exposure are fundamental parameters to be taken into account before drawing misleading conclusions. As side note, in fact, group II which is the one starting ART later also displays a better anti-HIV response due probably to prolonged antigen exposure.

AK Response: Lines 191-193 in the original submission (now lines 200-203 in the revised submission) very clearly state that “The failure to detect Gag-specific T cell responses in cohort 1 are also consistent with numerous reports that demonstrate a diminution or quiescence of the functional T cell response in

the face of prolonged viral suppression following early ART initiation^{7,8,34-44}. Additionally, lines 325-330 in the original submission (now **lines 317-322 in the revised submission**) also state that “Several studies have examined T cell responses in HIV+ pediatric subjects, and overall our data are concordant with an overwhelming majority of these reports that have demonstrated that early ART initiation and sustained virologic suppression in HIV-infected children can attenuate the virus-specific T cell response (which is clearly induced and detectable at birth in most infected infants), suggesting a link between measurable viremia and maintenance of a detectable virus-specific immune response^{7,8,34-44,62,63}”.

4- Childhood vaccines-induced serum Ab responses. Authors need to clearly indicate timing of last received vaccination to validate these results.

AK Response: Figure 7 already includes summary data indicating the interval between latest vaccine dose and the serum Ab assessment for this study. We are also including additional information related to this point in Supplementary Figure 8.

5- Cytokine production. It is important here to look at the baseline unstimulated levels in group I, II and HC. Delayed ART-start (group II), can be associated with basal higher levels of immune activation, cytokine production et.. Thus, higher level in this case may indicate a pro-inflammatory state consistent with previous literature, rather than suggest an improved functionality.

AK Response: The legends for figures 2, 5 and supplementary Figure 2 in the original submission (as well as this revised submission) clearly state that the numeric values calculated and depicted represent background-subtracted frequencies. The background was determined from both unstimulated samples stained with the antibodies for IFN γ , TNF α , IL-2, IL-21 and CD107a, as well as samples stimulated with either PMA/Ion or the HIV Gag PTE pool and stained with the relevant and dose-matched isotype control Ab for IFN γ , TNF α , IL-2, IL-21 and CD107a. Hence, what we are observing is indeed better functionality in cohort 2.

6- The authors should comment on the fact that starting ART-later (group II) is also associated with poor virological control (supp table 1), in line with previous literature.

AK Response: The difference in the day of life when ART was initiated in the two cohorts is **not** statistically significant ($p=0.3939$, Mann Whitney test). Please refer to the revised Supplementary Table 1 (row depicting Age at ART initiation). Although it is possible that this difference might be contributing to some extent to the sub-optimal virologic control in Cohort 2, we believe that the greater inconsistency in adherence to ART in cohort 2 is actually the primary contributor to the poor virologic control and selection of ART resistant virus strains in this group. This point has been added to the text (**lines 136-139**).

Reviewers' comments:

Reviewer #1 (Remarks to the Author):

The manuscript has been improved, I have no more comments.

Reviewer #2 (Remarks to the Author):

Overall, the authors have addressed my point in the revised manuscript.

Clarifications/concerns:

1- Numerosity of the samples: whilst I acknowledge that PHIV is USA are rare, it is important to critically discuss the fact the the presented results are obtained in a small sample group, this is a limitation of this study and these results may not be applicable to larger cohorts

2-Time of ART initiation: I am satisfied with the additions.

3- Cytokine production: regardless to the experimental controls that are in place (background-subtraction), my point is about the physiology of group II. The figure 2 is highly processed, and it hard to understand if the baseline levels of the cytokines in group II is driving the results. Have the authors measured also plasma cytokines in these 2 groups?

We thank the reviewers for their comments. A point-by-point response to the reviewer comments is provided below. The revisions in the main manuscript file are highlighted in blue.

Reviewers' comments:

Reviewer #1 (Remarks to the Author):

The manuscript has been improved, I have no more comments.

AK Response: Thank you.

Reviewer #2 (Remarks to the Author):

Overall, the authors have addressed my point in the revised manuscript.

Clarifications/concerns:

1- Numerosity of the samples: whilst I acknowledge that PHIV in USA are rare, it is important to critically discuss the fact that the presented results are obtained in a small sample group, this is a limitation of this study and these results may not be applicable to larger cohorts.

AK Response: Done as requested. Please refer to lines 401-406 in the revised manuscript.

2-Time of ART initiation: I am satisfied with the additions.

AK Response: Thank you.

3- Cytokine production: regardless to the experimental controls that are in place (background-subtraction), my point is about the physiology of group II. The figure 2 is highly processed, and it is hard to understand if the baseline levels of the cytokines in group II is driving the results. Have the authors measured also plasma cytokines in these 2 groups?

AK Response: We have now added the baseline levels of cytokine production in Supplementary Table 3 (this table is also now referenced in the Methods section: line 449). The listed values are **not** statistically significant between and among the groups. We have not yet measured plasma cytokines in our study cohorts. In order for the plasma cytokine level data to possess the same degree of resolution as the intracellular cytokine data as reported in this study, it would require measuring plasma cytokine levels using cell-sorted lymphocyte populations treated with the relevant stimulants. Given this, we believe that the intracellular cytokine data provides us with the necessary depth and breadth of data required to assess background and stimulated levels of cytokine production.

REVIEWERS' COMMENTS:

Reviewer #2 (Remarks to the Author):

I am happy with the new revision, the paper is ready for publication